# Synthesis and Application in Cell Imaging of Acridone Derivatives

**Yung-Chieh Chan** [1], **Chia-Ying Li** [2,3], **Chin-Wei Lai** [4] , **Min-Wei Wu** [4], **Hao-Jui Tseng** [4] and **Cheng-Chung Chang** [1,4,*]

1 Intelligent Minimally-Invasive Device Center, National Chung Hsing University, Taichung 402, Taiwan; yungchieh.c@gmail.com
2 Department of Surgery, Show Chwan Memorial Hospital, Changhua City 500, Taiwan; b86401115@ntu.edu.tw
3 Tissue Engineering and Regenerative Medicine, National Chung-Hsing University, Taichung 402, Taiwan
4 Graduate Institute of Biomedical Engineering, National Chung Hsing University, Taichung 402, Taiwan; alenpen5098@gmail.com (C.-W.L.); miss212518@gmail.com (M.-W.W.); horyzeng@gmail.com (H.-J.T.)
* Correspondence: ccchang555@dragon.nchu.edu.tw

**Abstract:** Tricyclic acridone derivatives have been extensively developed as antimicrobial, antimalarial, and antitumor drugs due to their broad spectrum of drug design and biological activity. In this study, we developed a surfactant-like acridone scaffold that contained two vinylpyridines and a dodecyl pyridine chain. The acridone scaffold decorated the dodecyl pyridine chain by N-bromosuccinimide reagent. The surfactant-like core scaffold incorporated with 4-vinylpyridines at the 2- and 7-positions via a Heck coupling reaction. Subsequently, the acridone derivatives were methylated onto these pyridine groups. Here we developed two similar acridone derivatives, MedAcd12C and MedAcd12P. The MedAcd12C incorporated two pyridine groups, and MedAcd12P incorporated three pyridine groups. MedAcd12C and MedAcd12P have two identical vinylpyridines and the different anchor tails at the N10 position. Their physicochemical properties, cell compatibility, and photoluminescence were demonstrated. Although both compounds have no fluorescence emission in water solution, MedAcd12P and MedAcd12C significantly appeared with orange light emission in the cellular imaging. We suggested that the surfactant-like scaffold promoted the drugs' self-assembly and caused the aggregation-induced emission (AIE) after cellular uptake. This innovative design endowed acridone derivatives with an AIE and traceability for cell imaging.

**Keywords:** acridone; fluorophore; aggregation-induced emission enhancement; fluorescent organic nanoparticles; fluorogen; biomarker

## 1. Introduction

Acridone is a well-known core structure in natural products, the derivatives of which are observed in various pharmacological applications [1,2]. Many as-such acridone derivatives are extracted due to their use as natural drug efficacy. Glyfoline derived from *Glycosmis citrifolia* was found to be a potent antineoplastic drug due to the inhibition of nucleoside transport function and injury of the G2/M phase of the tumor cells [3,4]. Acronycine, a natural acridone alkaloid isolated from *Acronychia baueri*, has significant anticancer activities as a result of its interference in DNA replication of tumor cells [5,6]. A potent anticancer drug, C1311, which is in progress in clinical trials, can bind tumor DNA to inhibit the proliferation of cancer cells [7]. Moreover, several artificial acridone derivatives have acquired the antibacterial and anticancer abilities by incorporating an alkyl chain onto the N-position, such as with pyrimidocarvazones or pyrimidoacridone, respectively [8,9]. Studies on acridone derivatives shows promising efficacy towards the development of potent anticancer drugs.

Acridone has an electron-deficient aromatic scaffold, which incorporates a π-conjugated electron donor to form a donor–acceptor chromophore [10]. Based on such tricyclic scaffolds that are susceptible to chemical modification, acridone derivatives have a range of versatile bioactivities and unique optical properties. Moreover, fluorescence from this organic molecule is normally quenched under aqueous conditions. This is due to the natural aggregation of the intrinsic hydrophobic aromatic structures that leads to an aggregation-caused quenching [11]. Given that an adaptable fluorophore normally incorporates various ionic groups, the issues to be addressed are that most donor-acceptor chromophores quench their fluorescence due to the intramolecular charge transfer (ICT) or twisted intramolecular charge transfer (TICT) in aqueous solvents [12–14]. Fluorophores exhibit aggregation-induced emission (AIE) depending on their particular structures [15–17]. Unlike chromophores with ICT or TICT above, the AIE luminogens (AIEgens) reveal brilliant fluorescence in aqueous conditions [18,19]. Recently, the AIEgens have been extensively studied and applied in cellular imaging and various biomedical applications, such as organic molecular frameworks, photosensitive liposomes, AIEgen-encapsulated nanoparticles, and image-guided drug delivery systems [20–23]. Thus, the adaptable acridone could be properly designed to form an AIEgen in cellular imaging.

In this study, we prepared two acridone derivatives designed with a surfactant-like acridone scaffold, enabling an auto-assembly in aqueous media. The N10 position was incorporated with a dodecyl group or dodecylpiperidine, respectively. Subsequently, both the two compounds incorporated two 4-vinylpyridines into the 2,7-position via a Heck coupling reaction and methylation. These decorated piperidines serve as electron acceptors and rotational moieties in TICT processes. We observed the unique optical properties owing to the AIE phenomenon which in turn restricts TICT, selectively retrieving fluorescence emission in cellular imaging. Therefore, these surfactant-like acridone derivatives prove to be potential candidates in the development of fluorescence probes and demonstrate usage in various bioapplications.

## 2. Materials and Methods

### 2.1. Chemicals

All chemicals were of reagent grade. Dichloromethane (DCM), ethyl acetate (EA), hexane, acetone, toluene, tetrahydrofuran (THF), palladium(II) acetate, methanol (MeOH), ethanol (EtOH), dimethylformamide (DMF), dimethyl sulfoxide (DMSO), acetonitrile (MeCN), triethylamine (TEA) hexane, acetone (Ace), toluene, tetrahydrofuran (THF), and dimethylformamide (DMF) were purchased from Sigma-Aldrich. 9(10H)-Acridine, 1,12-dibromododecane, 1,4-dibromobutane, 1-bromododecane, 4-vinylpyridine, acridine, benzyltrimethylammonium tribromide, iodomethane, N-bromosuccinimide, n-butylamine were purchased from Acros. 1-Bromobutane was purchased from Alfa Aesar. Magnesium sulfate was purchased from Showa. Distilled hexane and acetone were used. Thin-layer chromatography (TLC) was purchased from Merck. Column chromatography, using Silica gel 60 F, 230–240 mesh ATSM were purchased from Merck. The solvent used in the spectroscopy is of HPLC grade and was purified following standard procedures.

### 2.2. Synthesis and Identification of MeAcd12C and MeAcd12P

Similar synthesis procedure for MeAcd12C and MeAcd12P incorporating 1-bromododecane and 1,12-dibromododecane was adopted. A total of 5.13 mmol of acridone, 5.14 mmol of potassium iodide, 9.09 mmol of potassium hydroxide, and 20 mL of THF were mixed in a pot and heated to 40 °C for 1 h. 10-(12-bromododecyl) acridone was extracted by the mixture of DCM and NaOH.

NMR spectroscopy was used to identify the compounds and intermediates present during the synthesis procedure (Figure S1a). Details of the [1]H-NMR data of MeAcd12P are as follows: [1]H NMR (400 MHz, DMSO-$d_6$): δ 8.87 (d, *J* = 6.5 Hz, 4H), 8.70 (d, *J* = 2.2 Hz, 2H), 8.28–8.24 (m, 6H), 8.22 (d, *J* = 10.9 Hz, 2H), 8.00 (d, *J* = 9.2 Hz, 2H), 7.62 (d, *J* = 16.3 Hz, 2H), 4.58 (s, 2H), 4.26 (s, 6H), 3.27 (dd, *J* = 11.2, 2.1 Hz, 4H), 2.96 (s, 3H), 1.83 (s, 2H), 1.78–1.70 (m, 4H), 1.63 (s, 2H), 1.58–1.44 (m, 4H),

1.38 (s, 2H), 1.28 (s, 14H). MS(ESI,m/z): 652.4; found, 654.6; Anal. Calcd % for $C_{44}H_{52}N_4O$: C, 80.94; H, 8.03; N, 8.58; O, 2.45. Data for MeAcd12C (Figure S1b): $^1$H NMR (400 MHz, DMSO-$d_6$): δ 8.89–8.84 (m, 4H), 8.68 (d, *J* = 2.2 Hz, 2H), 8.25 (d, *J* = 1.5 Hz, 4H), 8.22 (d, *J* = 16.9 Hz, 2H), 8.00 (d, *J* = 9.3 Hz, 2H), 7.61 (d, *J* = 16.3 Hz, 2H), 4.58 (t, *J* = 7.7 Hz, 2H), 4.25 (s, 6H), 1.81 (d, *J* = 7.3 Hz, 2H), 1.51 (d, *J* = 7.9 Hz, 2H), 1.35 (d, *J* = 8.5 Hz, 2H), 1.24 (d, *J* = 10.3 Hz, 14H), 0.85–0.79 (m, 3H). MS(ESI,m/z): 569.3; found, 569.5; Anal. Calcd % for $C_{39}H_{43}N_3O$: C, 82.21; H, 7.61; N, 7.37; O, 2.81.

## *2.3. Apparatus*

Varian Mercury 400 and Varian Mercury plus 400 nuclear magnetic resonance spectrometers were used. The sample to be tested was dissolved in deuterated solvent ($CDCl_3$, DMSO-d6), and the structure and purity were judged by hydrogen nuclear magnetic resonance spectroscopy (1H-NMR) and carbon nuclear magnetic resonance spectroscopy (13C-NMR). The chemical shifts (δ) are expressed in ppm, the coupling constants are described as J values, and the unit is Hz, and the chemical shift of the solvent is used as the standard: the peak of $CDCl_3$ in 1H-NMR spectrum is δ = 7.26 ppm, the peak value of DMSO-d6 is δ = 2.49 ppm. Splitting is defined as follows: s, singlet; d, doublet; t, triplet; q, quartet; p, quintet; dd, doublet of doublets; m, multiplet. The absorption spectrum of chemicals was measured by UV-Visible spectrophotometer (Thermo Genesys 6, New York, NY, USA). Fluorescence spectra were obtained with a fluorescence spectrophotometer (Horiba Jobin-Yvon, Paris, France). Fluorescence images were acquired with a Leica AF6000 fluorescence microscope with a DFC310 FX digital color camera.

## *2.4. Characterization*

Optical properties were studied by absorption and fluorescence spectrometers, respectively. All glass cuvettes were cleaned several times with acetone and double-deionized water (DDW), and then air-dried before their first use. The tested acridone derivatives were prepared at a concentration of $10^{-5}$ M in various solvents, such as DMSO, methanol, ethanol, acetonitrile, DDW, acetone, ethyl ethanoate, tetrahydrofuran, dichloromethane, and dimethylformamide, respectively. Optical properties of these samples were measured by absorbance and fluorescence spectrophotometers, respectively. The sizes distribution of AIEgen were determined by dynamic light scattering analysis (nanoPartica SZ-100V2, HORIBA, Kyoto, Japan).

## *2.5. Determination of Quantum Yield*

The quantum yield of acridone derivatives was determined according to the following equation:

$$\Phi_u = \Phi_S \times [A_{fu} \times A_s(\lambda_{exs}) \times \eta_u^2]/[Afs \times A_u(\lambda_{exu}) \times \eta_s^2]$$

The $\Phi_u$ is the quantum yield of the compound, $A_f$ is the integrated area under the emission spectra; $A(\lambda_{ex})$ is the absorbance area at the excitation wavelength; η is the refractive index of the solution; the subscripts s and u indicate the unknown and standard, respectively [24]. According to our previous studies, we chose 3,6-bis(1methyl-4-vinylpyridium) carbazole diiodide as the standard subject, which showed a quantum yield of 0.02 in DMSO and 0.25 in glycerol [25].

## *2.6. Cell Culture and Cellular Imaging*

The human cervical cancer cell line, HeLa and nontumorigenic human breast epithelial cell line, H184B5F4/M10 were cultured in MEM supplemented with penicillin-streptomycin-glutamine (GIBCO; final concentration: 100 units of penicillin, 100 µg streptomycin, and 0.292 mg glutamine per ml medium) and 10% FBS (Invitrogen) [26]. All cells were cultured and incubated in a 37 °C incubator containing 5% $CO_2$. All cell lines were purchased from the Bioresource Collection and Research Center (BCRC), Taiwan. Cells were seeded onto coverslips and treated with compounds for 12 h. For fluorescence imaging, cells were first washed by PBS and then transferred onto glass slides.

## 3. Results

### 3.1. Subsection

#### 3.1.1. Molecular Design

The synthetic procedure for MeAcd12C incorporating two 4-vinylpiperidines and a 1-bromododecane was depicted in Figure 1a. 1-bromododecane was incorporated into the N10 position of the acridone, and was then boronated by NBS reagent. The Heck cross coupling reaction and methylation were followed to acquire the final product. All compounds were identified by routine spectral and elemental analyses. The synthetic procedure of the dodecyl-piperidine-substituted acridone (MeAcd12P) was similar to that of MeAcd12C. 1,12-dibromododecane was incorporated into the N10 position of acridone, where the twelve-carbon chain exhibited lipophilic properties (Figure 1b). The bromo-substituted end of dodecyl chain was replaced with piperidine to form dodecyl-piperidine acridone intermediate. This dodecyl-piperidine acridone scaffold was brominated at positions 2 and 7 by N-bromosuccinimide (NBS). Here, bromination of the acridone scaffold was facilitated by NBS because of the incorporated electron-donating group at the N10-position of acridone. Consequently, the 2,7-dibromo-N-(dodecyl) acridone scaffold was incorporated with the two 4-vinylpyridines via a Heck cross-coupling reaction, and methylated with iodomethane to obtain the final product, MeAcd12P.

**Figure 1.** Schemes representing synthesis procedures and the corresponding yields from (**a**) MeAcd12C and (**b**) MeAcd12P conditions; (**a**) i, 1-bromododecane, KI, KOH, THF, 40 °C, 62%; ii, NBS, DMF, 80 °C, 83.7%; iii, 4-vinylpyridine, Pd(OAc)2, (o-tol)3P, MeCN, Et3N, N2, 80 °C, 30%; iv, CH3I, DMF, Acetone, 80%. (**b**) i, 1,12-dibromododecane, KI, KOH, THF, 40oC, 54%; ii, NBS, MeCN, reflux, 97%; iii, piperidine, K2CO3, THF/H2O, reflux, 91.6%; iv, 4-vinylpyridine, Pd(OAc)2, (o-tol)3P, MeCN, Et3N, N2, 80 °C, 63%; v, CH3I, DMF, Acetone, 80%.

### 3.1.2. Photophysical Properties of MeAcd12C and MeAcd12P

MeAcd12C displays two distinct absorption bands, the major peak appearing at 420–490 nm and the minor peak at 325–350 nm, respectively (Figure 2a). When MeAcd12C is dissolved in protic organic solvents such as DMSO, MeOH, EtOH, MeCN, and Ace, the major absorbance peak appears at 450 nm. In $H_2O$, MeAcd12C exhibited blue-shifted absorbance when compared to other protic organic solvents. However, an increasing aprotic environment, including DCM and THF, resulted in a red shift in the absorbance of the MeAcd12C spectrum. The fluorescence spectrum from MeAcd12C displayed emission wavelengths between 560–590 nm (Figure 2b and Figure S2a) and exhibited lower fluorescence intensities in $H_2O$, EA, and THF. A similar solvent effect occurred for MeAcd12P, which displayed two distinct absorption spectra (Figure 2c). The major absorption peak of MeAcd12P in protic organic solvents appeared at 450 nm. The absorbance of MeAcd12P in aprotic solvent exhibited a larger red-shift than that of MeAcd12C, thus implying that the additional dodecyl piperidine might cause this shift. The fluorescence emission intensities from MeAcd12P were slightly weaker than those from MeAcd12C (Figure 2c and Figure S2b). Moreover, MeAcd12P has a lower-fluorescence emission in $H_2O$, DCM THF, and EA, thus implying the presence of TICT.

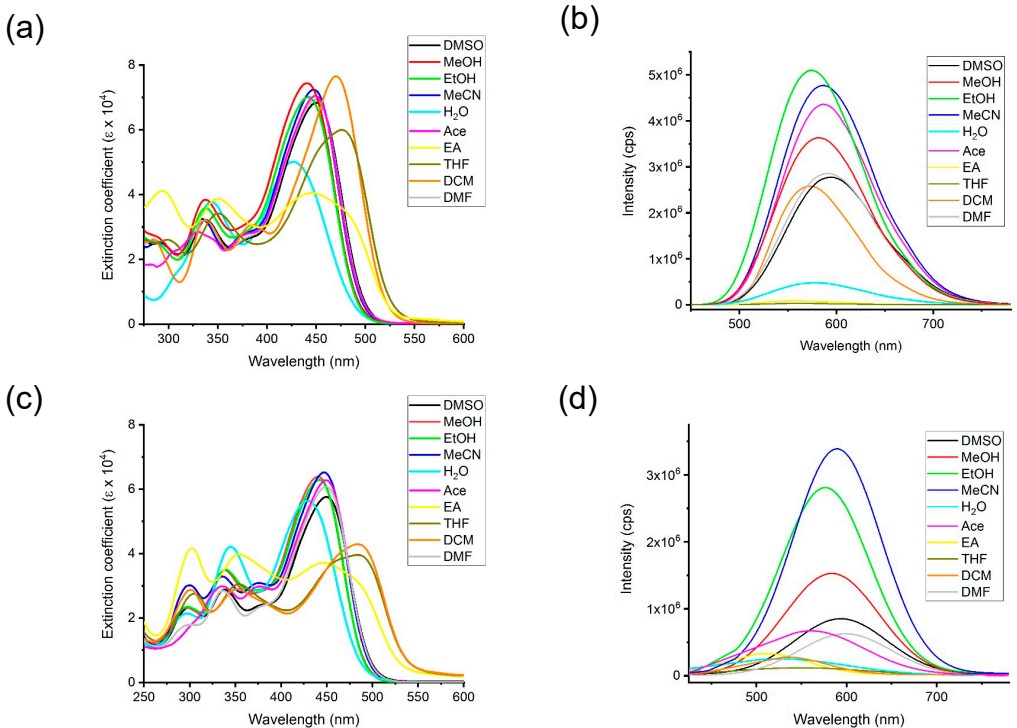

**Figure 2.** Normalized absorbance and fluorescence emission spectra of MeAcd12C (**a**,**b**) and MeAcd12P (**c**,**d**) in different solvents. The corresponding excitation wavelengths were determined from the absorption maxima present in each absorbance spectrum.

### 3.1.3. Optical Properties of MeAcd12C and MeAcd12P

The orientation polarizability of solvents, maximum wavelength of absorption and emission, Stokes shift, and quantum yield are listed in Table S1. The absorbance shifts are related to the solvent polarities for MeAcd12C and MedAcd12P. Moreover, the solvent sensitivity was determined by the Lippert plots according to the absorption and emission spectra from MeAcd12C and MedAcd12P in various solvents [27]. The Stokes shifts from MeAcd12C and MedAcd12P were evaluated by the Lippert–Mataga equation, which is related to the energy difference between the ground and excited states (Figure S3, Table S1). The orientation polarizability values for each solvent compared with the corresponding Stokes shifts are shown in Figure 3, where the slope indicates the solvent sensitivity

value from MeAcd12C and MedAcd12P. Positive solvatochromism was found in the solvent polarity dependence, thus implying that the two compounds are intramolecular charge transfer (ICT) emissive states [28]. The Stokes shift of MeAcd12C was dependent on the polarity values, but with a degree of deviation from the slope in the polar aprotic solvents such as EA and THF (Figure 3a). The fluorescence quantum yield from MeAcd12C appeared to be correlated to the solvent polarity except (1.02%). It is interesting to observe the unexpectedly reduced quantum yields and the blue-shift absorbance in $H_2O$. Although the slope of Lippert–Mataga plot was similar to that of MeAcd12C, MeAcd12P showed convergence in EA and THF (Figure 3b). The Stokes shift of MeAcd12P was significantly reduced and deviated from the slope in $H_2O$ and DCM. The quantum yield from MeAcd12P was relatively smaller than that from MeAcd12C. Moreover, there is little fluorescence change of the two compounds, indicating that photostability of MeAcd12C and MeAcd12C was stable (Figure S4).

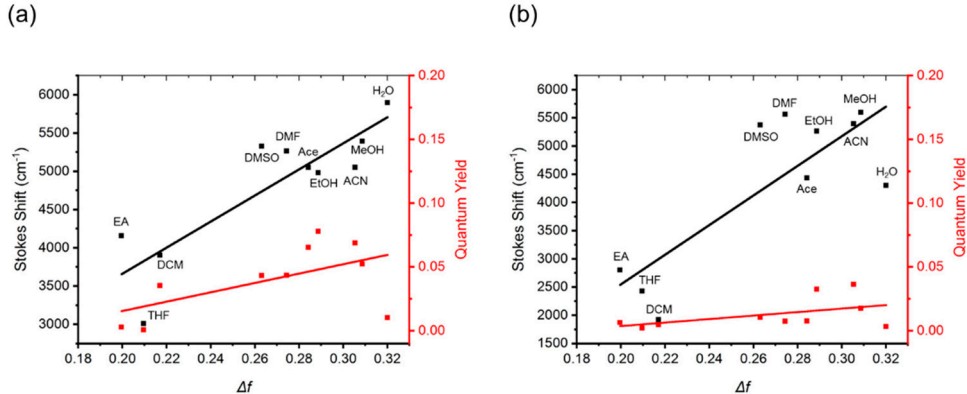

**Figure 3.** The Lippert–Mataga plots and the corresponding quantum yields of MeAcd12C (**a**) and MeAcd12P (**b**). The values of the Stokes shifts were calculated from the observed maxima present in the absorption and emission spectra, respectively. Values of the quantum yields were determined by a comparative method.

### 3.1.4. Systemic Cytotoxicity

Both H184B5F5/M10 and HeLa cell lines were susceptible with MeAcd12C and MeAcd12P (Figure 4a,b). Over 90% of both cells survived after treatment under concentrations of 1.0 μM. However, a divergence of viability appeared at 1.0 μM, in which about 50% of HeLa cells were inhibited by MeAcd12P. Comparatively, 50% of HeLa were inhibited by 3.0 μM of MeAcd12C treatment. In total, 80% of H184B5F5/M10 cells were alive before the 3.0 μM of both compounds. Additionally, 50% of H184B5F5/M10 cells were inhibited by 10.0 μM of MeAcd12C. The half maximal effective concentration (EC50) of MeAcd12C showed 3.0 μM for HeLa cells and 10.0 μM for H184B5F5/M10 cells. The EC50 of MeAcd12P was at 1.0 μM for HeLa cells. This indicated that MeAcd12P might exhibit selective cell toxicity compared to MeAcd12C.

### 3.1.5. Cellular Imaging Application

Both H184B5F5/M10 and HeLa cells were incubated with MeAcd12C and MedAcd12P for 24 h. This was done in order to examine whether the entry and intracellular localization of these MeAcd12C and MedAcd12P could be used as fluorescent probes. The yellow-orange speckles from MeAcd12C were observed from both the cell lines (Figure 5a). Intracellular imaging of MeAcd12P showed similar behavior to that of MeAcd12C in H184B5F5/M10 cells (Figure 5b). MeAcd12P appeared as orange speckles and preferred to accumulate closer to the nucleus. However, the uptake of MeAcd12P in HeLa did not present specific localization.

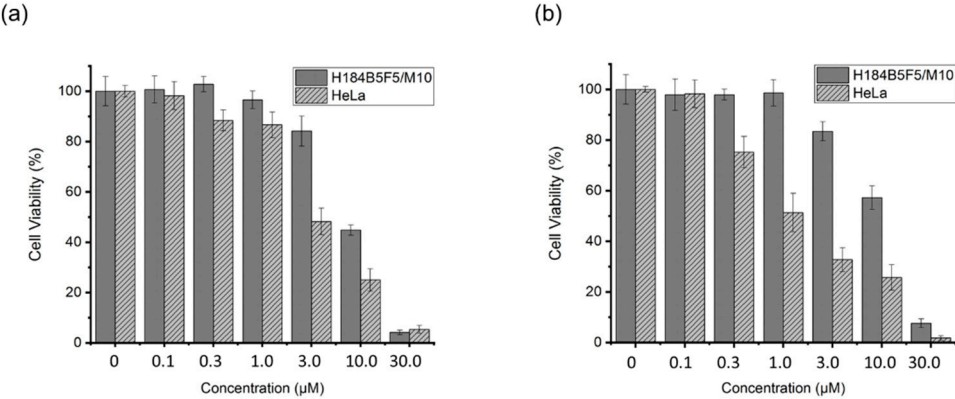

**Figure 4.** Normalized cell viability of Hela and H184B5F5/M10 cells treated with various concentrations of MeAcd12C (**a**) and MeAcd12P (**b**). Compounds were prepared with concentration values of 0, 0.1, 0.3, 1.0, 3.0, 10.0, and 30.0 μM, respectively. After 24 h incubation, trypan blue staining was used to calculate cell viability. Each data represented three measurements with the calculated standard error.

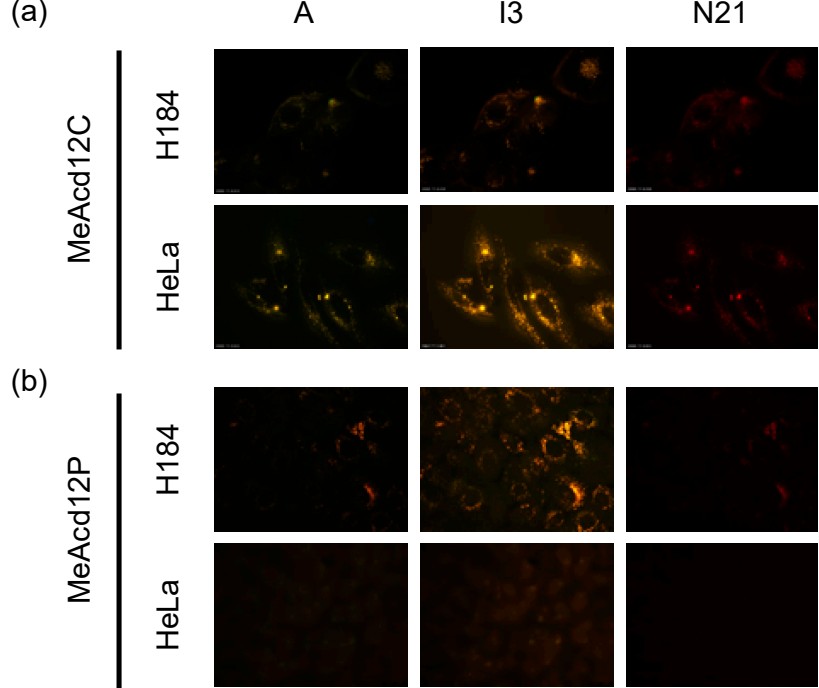

**Figure 5.** Fluorescent images of Hela and H184B5F5/M10 (H184) cells treated with 1 μM of MeAcd12C (**a**) and MeAcd12P (**b**) after 24 h of incubation. A: ex, 350; em; 425, long pass filter (lp). I: ex, 450; em; 515, long pass filter (lp). N: ex, 510; em; 590, long pass filter (lp).

3.1.6. Investigation of Cellular Imaging Application

We observed that both MeAcd12C and MedAcd12P appeared fluorescent in the mixed solvents (Figure S5). This indicates that these surfactant-like compounds would autoassembled to form AIEgens due to their amphiphilic structures. Both the AIEgens MeAcd12C and MedAcd12P could be examined by fluorescence microscopy (Figure S5c,d). The sizes of the MeAcd12C fluorogens were approximately 500 nm, and those of the MeAcd12P fluorogens were approximately 400 nm. The formation of fluorogens indicated that the TICT of the two compounds reduced under the mixed solvent conditions. Moreover, the major AIEgen sizes of MeAcd12C and MedAcd12P were 250 and 150 nm in $H_2O$/THF

mixed solvents ($H_2O$/THF = 50%:50%), respectively (Figure S6a,b). These indicated that fluorescence of MeAcd12C and MeAcd12P appeared due to the AIE formation.

The primary optical spectra from MeAcd12C and MedAcd12P in $H_2O$/THF mixed solvents were examined. Absorption spectra from the MeAcd12C fluorogens displayed similar patterns in the polar aprotic solvent as well (Figure 6a). The volume fraction of 50% $H_2O$ exhibited the highest fluorescence emission (Figure 6b), thereby indicating that a substantial proportion could produce MeAcd12C fluorogens. Similarly, the absorbance of MedAcd12P fluorogens matched with its primary basic spectra, with the major absorbance peaks appearing near 450 nm (Figure 6c). The appropriate volume of $H_2O$ to produce MedAcd12P fluorogens was at a range of 25–50%, which yielded high fluorescence emission intensities (Figure 6d). The AIEgen spectra of MedAcd12C and MedAcd12P were similar to the spectra of their solid-state emission (Figure S7). We also found that fluorescence intensity of MeAcd12C reduced in the present of BSA and DNA (Figure S8). However, there is little fluorescence change when the MeAcd12P met BSA and DNA. The results implied that the increasing fluorescence of the MeAcd12C and MeAcd12P was irrelevant to the existence of BSA and DNA in cells.

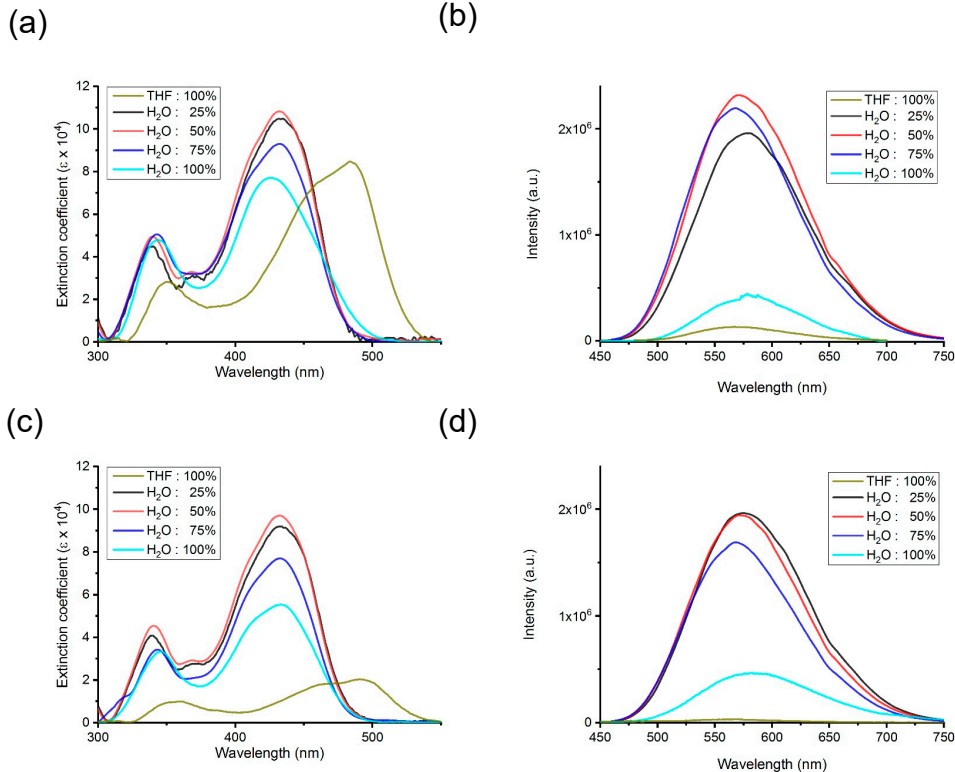

**Figure 6.** Normalized absorption and fluorescence emission spectra of fluorogens from MeAcd12C (**a**,**b**) and MeAcd12P (**c**,**d**). Compounds were dissolved in 0%, 25%, 50%, 75%, and 100% of $H_2O$ mixing ratios with THF (*v/v*). The corresponding excitation wavelengths were determined from the absorption maxima present in each absorbance spectrum.

## 4. Discussion

In this study, we synthesized the biocompatible acridone derivatives, comprising an electron acceptor vinyl pyridinium and an electron donor acridone. Acridone is known as a biocompatible and media-sensitive chromophore, the scaffold of which could easily be combined with other functional groups to form unique biomarkers. In order to increase the optical diversity, the acridone core was conjugated with vinylpyridine groups on the 2,7-positions to form a donor–acceptor chromophore, leading to the twisted intramolecular charge transfer (TICT) property. Furthermore, a dodecyl group was incorporated into the N10 position, named MeAcd12C, which serves specific targeting ability for

cell imaging. We also replaced the dodecyl group with N-dodecyl-N-methyl-piperidinium to obtain MeAcd12P, which was further characterized to reveal its physicochemical properties and was subjected to different applications.

By considering acridone as a major electron donor in this study, we synthesized MeAcd12C and MedAcd12P decorated with the distinct electron acceptors. Although two absorption peaks from MeAcd12C, MedAcd12P, and acridone could be observed, the basic optical spectra of MeAcd12C and MedAcd12P were significantly affected by the solvent polarity [29]. From the optical properties as observed in our study, we suggested that the optical diversities could be due to the incorporated vinyl pyridinium and dodecyl pyridinium, respectively, which were employed as electron acceptors. The major absorption peaks from the two compounds located at 450 nm might originate from the acridone core. Although there were three absorption peaks from MeAcd12C and MedAcd12P in EA, no emission peaks from these compounds could be observed. A hypochromic shift was observed with increasing solvent polarity. It is important to note that the fluorescence intensities from MeAcd12C and MedAcd12P were observed to be relatively lower in magnitude while in $H_2O$ than in any other protic organic solvent. This suggests that the two acridone derivatives possess selective fluorescence depending on the environment, which proved to be advantageous in cell imaging.

We observed that the quantum yields of the two compounds were independent of solvent polarity. The two compounds exhibited high quantum yields in polar protic organic solvents, such as MeOH and EtOH, and displayed yellow fluorescence. Polar aprotic solvents such as DMSO, MeCN, Ace, and DMF resulted in a red-shifted fluorescent emission. However, very low quantum yields were obtained in $H_2O$, THF, and DCM. The selective photoluminescence abilities of MeAcd12C and MedAcd12P could be exploited as potential biomarkers in cell imaging.

Acridone has been extensively studied in the induction of cell apoptosis and in the development of chemotherapeutic agents [30,31]. Treatment with acridone inhibits the ATP-binding cassette subfamily G member 2 protein to improve the multidrug resistance in breast cancer therapy [32,33]. In order to evaluate the biocompatibility of MeAcd12C and MedAcd12P, we studied their cytotoxicity behavior in the nontumorigenic H184B5F5/M10 and cancerous HeLa cell lines at various concentrations. The EC50 value of MeAcd12P was approximately at 1.0 μM for HeLa cells and in the range of 10–30 μM for H184B5F5/M10 cells. Both MeAcd12C and MedAcd12P showed selective cell toxicity for H184B5F5/M10 and HeLa cells, thus implying that the acridone scaffold derivatives are potential candidates in cancer therapy. We aim to further investigate the intrinsic cytotoxicity effects in other cell lines to understand the underlying mechanisms.

In addition to studies on optical properties and biocompatibility issues, MeAcd12C and MedAcd12P were further subjected to cellular accumulation and bioimaging analysis. It was observed that the MeAcd12C speckles made their significant appearances surrounding the H184B5F5/M10 nucleus compared to the localization in HeLa cells. Moreover, MeAcd12C was distributed in the cytoplasm of the HeLa cells, and the appearance of the unexpectedly strong speckles might be due to aggregation. However, the specific localization of MeAcd12P in HeLa cells were not very prominent even under the conditions of reduced concentrations and incubation times. We were particularly interested in the cellular imaging of MeAcd12P in H184B5F5/M10 and HeLa cells. Considering the cell viability after treatment with the two compounds, MeAcd12P was demonstrated to possess specific killing ability towards HeLa cells than over H184B5F5/M10 cells. Cellular staining tests showed the two compounds to be uptaken by the cells and remained in the cytoplasm. Understanding both the fluorescence mechanism and subcellular localization effects are topics worthy of future investigation.

Although MeAcd12C and MedAcd12P were observed to enter cells and showed fluorescent speckles in the cytoplasm, the results did not coincide well with the optical properties, which presented very low quantum yield in $H_2O$. Considering that the two compounds were designed as surfactant-like chromophores along with the presence of fluorescent speckles, we further investigated whether the fluorescent speckles were due to the aggregation-induced emission (AIE) effects [34,35]. The AIE effects on MeAcd12C and MedAcd12P were examined under varying $H_2O$/THF mixed solvent system ratios

(Figure S5). Therefore, the fluorescence enhancement of the two compounds was dependent on the AIE phenomenon but their emission wavelengths were irrelevant to the formation of fluorogens.

## 5. Conclusions

Two acridone derivatives, MeAcd12C and MedAcd12P, were synthesized and characterized to understand their specific optical behaviors. The optical absorption spectra displayed blue-shifts, which were found to be correlated to the increased values in the solvent polarity. Both the two compounds demonstrated stable fluorescence emission in the wavelength regions of 560 to 590 nm, except in $H_2O$ and THF. The presence of an additional dodecyl piperidine group in MedAcd12P could promote better TICT progress progression than MeAcd12C. Although the TICT progress resulted in reduced fluorescence intensity in $H_2O$, we observed the selective fluorescence effects to be prominent in cellular imaging due to the AIE effects. These unique optical behaviors observed in the two compounds envisage the two acridone derivatives as potential fluorescent probes for use in various future bioapplications.

## 6. Patents

This section is not mandatory, but may be added if there are patents resulting from the work reported in this manuscript.

**Supplementary Materials:** The following are available online at http://www.mdpi.com/2076-3417/10/23/8708/s1. Figure S1: The NMR spectra of MeAcd12C and MeACD12P. Figure S2: Photograph of MeAcd12C and MeAcd12P corresponded to absorption and fluorescence emission spectra. Figure S3: The Lippert-Mataga equations. Figure S4: Photostability of MeAcd12C and MeAcd12P. Figure S5: The Photograph of MeAcd12C and MeAcd12P in mixed solvent system ratios. Figure S6: Dynamic light scattering analysis. Figure S7: Solid state fluorescence emission spectra. Figure S8: Optical effects of MeAcd12C and MeAcd12P with BSA and DNA. Table S1: Photophysical information.

**Author Contributions:** C.-C.C. conceived the study. Y.-C.C., C.-Y.L. and C.-C.C. carried out experimental design. Y.-C.C., C.-W.L., M.-W.W. and H.-J.T. performed the experiments. C.-W.L. and C.-Y.L. analyzed the data. The manuscript was drafted by Y.-C.C., C.-Y.L., and edited by C.-C.C. All authors have read and agreed to the published version of the manuscript.

**Funding:** This research received no external funding.

**Acknowledgments:** The authors thank the Ministry of Science and Technology (MOST 108-2113-M-005-006 -) (MOST 109-2224-E-005-001 -) of Taiwan for financial support.

**Conflicts of Interest:** The authors declare no conflict of interest.

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
