# Peer review of "Synthesis and Application in Cell Imaging of Acridone Derivatives"

_applsci, doi:10.3390/app10238708_

Round 1
Reviewer 1 Report
This manuscript is basically a draft with full of spelling errors and missing requirements. There is no correct description of the syntheses, no NMR spectra attached, no high resolution MS or elemental analysis results. Many sentences stayed in the mansucript from the original template. The abstract presents too exact informations, particularly results, reactions, compound IDs. Cell viability assays might be correct, but there is no IC50 in these cases, and the descibed determination is not correct at all. Photostability would also be required.
Author Response
Thank reviewer 1 for the critical comments. We have carefully revised and hope the responses satisfying the raise comments. The NMR spectra of MeAcd12C and MeAcd12C have been added in supplementary (Fig. S1). The corresponding depiction of NMR and MS and elemental analysis results were revised (Page 2, line 112). We have carefully revised the manuscript and processed the English-Editing. The revised sentences were highlighted in yellow. We agreed that the current results of the Systemic cytotoxicity are insufficient for IC50 determination. The description has been revised (Page 5, line 212-218). The photostability of MeAcd12C and MeAcd12C have been confirmed that the time-lapse fluorescence spectra were acquired under the continuous irradiation (Fig. S4). There is little fluorescence change of the two compounds, indicated that photostability of MeAcd12C and MeAcd12C were stable (Page 5, line 247).
Reviewer 2 Report
Cheng-Chung Chang et. al reported work Synthesis and Applied in Cell Imaging of Acridone Derivatives is interesting can be acceptable upon major revision as specified below.
1) Authors need to specify the conditions of Quantum yield measurements such as standard and methods they have used in the experimental sections clearly.
2) Emission spectra were recorded at which excitation wavelength ? They have to mention it figure captions.
3) Authors need to check the optical effects of probes with few bio molecules such as DNA, BSA along with negatively charged surfactants in the invitro conditions in order to prove their AIE hypothesis.
4) Solid state emission spectra should be presented.
5) Authors need to check the DYNAMIC LIGHT SCATTERING METHOD to identify the aggregations in H2O conditions. By looking at the UV-Vis spectra, upon addition of H2O to THF solution, blue shift was observed. Hence this evidance can clearly provide a deep insights towards probes optical properties Either AIE or Rigidity induced fluorescence enhancement accompanied through the restricted C=C bond rotations.
6) Bioimaging studies not clearly presented with nucleus staining dye either DAPI/Hoechst 33342. Becuase Fig. 5a and 5b showed different cell localisation with different compounds. That means each compounds showed different localisations in different cell models. Then how authors conclude that cytosolic localiasation. This looks very ambiguous . Authors need to correct it with suitable explanations.
7) Illustrations and representations were not clear in results in discussion, authors need to stress the significant point in this work. There are few sentences seems to be rewritten in few pages such as bio imaging studies page 11, 12 and photo physical studies and many more, wherever it appropriate.
Author Response
Comments and Suggestions for Authors
Cheng-Chung Chang et. al reported work Synthesis and Applied in Cell Imaging of Acridone Derivatives is interesting can be acceptable upon major revision as specified below.
- Authors need to specify the conditions of Quantum yield measurements such as standard and methods they have used in the experimental sections clearly.
Response:
Thank the reviewer 2 for the kind comments. We have added the experimental section of Quantum yield measurements, including the standard, equation and corresponding references (Page 3, line 145).
2) Emission spectra were recorded at which excitation wavelength? They have to mention it figure captions.
Response: Thank the reviewer 2 for pointing out the missing. We have revised the figure legends of figure 2 and 6. The revised sentences were highlighted in yellow (Page 7 line 333 and Page 10, line 382).
3) Authors need to check the optical effects of probes with few bio molecules such as DNA, BSA along with negatively charged surfactants in the invitro conditions in order to prove their AIE hypothesis.
Response:
Thank reviewer 2 for the important comments. We have recruited the in vitro experiment in which the fluorescence spectra of the two compounds with BSA and DNA were determined. Fluorescence intensity of MeAcd12C reduced in the present of BSA and DNA (Fig. S8). Moreover, there is little fluorescence change when the MeAcd12P met BSA and DNA. The results implied that the increasing fluorescence of the MeAcd12P was irrelevant to the exitance of BSA and DNA in cells. The revised sentences were added in the context (Page 6 line 314).
4) Solid state emission spectra should be presented.
Response:
Thank reviewer 2 for the comments. The solid-state emission has been presented. The spectra corresponded to the AIEgen spectra of MedAcd12C and MedAcd12P, implied that the fluorescence was due to aggregations of the compounds (Fig. S7, Page6 line 313).
5) Authors need to check the DYNAMIC LIGHT SCATTERING METHOD to identify the aggregations in H2O conditions. By looking at the UV-Vis spectra, upon addition of H2O to THF solution, blue shift was observed. Hence this evidance can clearly provide a deep insights towards probes optical properties Either AIE or Rigidity induced fluorescence enhancement accompanied through the restricted C=C bond rotations.
Response:
Thank reviewer 2 for the critical comments. We have checked the size distribution of MeAcd12C and MeAcd12P in H2O conditions by dynamic light scattering analysis. However, we did not find the significant aggregation signals. We further identified MeAcd12C and MeAcd12P in H2O/THF mixed solvents (H2O/THF=50%:50%), which appeared the aggregations (Fig. S6). These indicated that fluorescence of MeAcd12C and MeAcd12P in cell model due to the AIE formation. The revised sentences were added in the context (Page 5, line 277).
6) Bioimaging studies not clearly presented with nucleus staining dye either DAPI/Hoechst 33342. Becuase Fig. 5a and 5b showed different cell localisation with different compounds. That means each compounds showed different localisations in different cell models. Then how authors conclude that cytosolic localiasation. This looks very ambiguous. Authors need to correct it with suitable explanations.
Response:
Thank reviewer 2 for the kind comments. According to the bright field photo, uptake of each compounds localized in the cytoplasm. However, the quality of expose was poor leading to the ambiguous result. We have adjusted the expose of photos to the identical condition. Moreover, we agreed that the two compounds showed different localizations. We will identify the localization in the future.
7) Illustrations and representations were not clear in results in discussion, authors need to stress the significant point in this work. There are few sentences seems to be rewritten in few pages such as bio imaging studies page 11, 12 and photo physical studies and many more, wherever it appropriate.
Response:
Thank reviewer 2 for the comments. We have carefully revised the manuscript according to reviewers’ comments. The version would stress the major work that the two acridone derivatives have unique optical properties, selective cell inhibition and cell imaging. Although the two compounds showed lower fluorescence emission in the aqueous phase, the significant fluorescence appeared due to aggregation-induced emission (AIE).
Round 2
Reviewer 1 Report
Authors answered the questions and comments to both reviewers. For the proper characterizatipon of the new chemical compound, the carbon NMR is also necessary, therefore I suggest to include the spectra to the supplememntary and the annotation to the experimental. I would also recommend the proper characterization of the other synthesized chemicals on Fig. 1 reaction sequence and adding all the reactions to the experimental in a reproducible format. In addition, when a reaction is shown, that is usually not a "Figure" but a "Scheme". Now, the manuscript is in much better shape, it could be accepted after these changes.
Reviewer 2 Report
Article can be acceptable in current form.